# Denosumab and Zoledronic Acid Differently Affect Circulating Immune Subsets: A Possible Role in the Onset of MRONJ

**DOI:** 10.3390/cells12202430

**Published:** 2023-10-11

**Authors:** Ilaria Roato, Lorenzo Pavone, Riccardo Pedraza, Ilaria Bosso, Giacomo Baima, Francesco Erovigni, Federico Mussano

**Affiliations:** 1Bone and Dental Bioengineering Laboratory, CIR-Dental School, Department of Surgical Sciences, University of Turin, Via Nizza 230, 10126 Turin, Italy; lorenzo.pavone@edu.unito.it (L.P.); riccardo.pedraza@unito.it (R.P.); giacomo.baima@unito.it (G.B.); federico.mussano@unito.it (F.M.); 2Institute of Sciences and Technologies for Sustainable Energy and Mobility, National Council of Research, 10135 Turin, Italy; 3DIMEAS, Politecnico di Torino, 10129 Turin, Italy; 4CIR-Dental School, Città della Scienza e della Salute, 10126 Turin, Italy; ilaria.bosso@unito.it (I.B.); francesco.erovigni@unito.it (F.E.)

**Keywords:** anti-resorptive drugs, denosumab, osteonecrosis, T cells, zoledronate

## Abstract

This work investigated whether the anti-resorptive drugs (ARDs) zoledronic acid (Zol) and denosumab (Dmab) affect differently the levels of circulating immune cell subsets, possibly predicting the risk of developing medication-related ONJ (MRONJ) during the first 18 months of treatment. Blood samples were collected from 10 bone metastatic breast cancer patients receiving cyclin inhibitors at 0, 6, 12, and 18 months from the beginning of Dmab or Zol treatment. Eight breast cancer patients already diagnosed with MRONJ and treated with cyclin inhibitors and ARDs were in the control group. PBMCs were isolated; the trend of circulating immune subsets during the ARD treatment was monitored, and 12 pro-inflammatory cytokines were analyzed in sera using flow cytometry. In Dmab-treated patients, activated T cells were stable or increased, as were the levels of IL-12, TNF-α, GM-CSF, IL-5, and IL-10, sustaining them. In Zol-treated patients, CD8+T cells decreased, and the level of IFN-γ was undetectable. γδT cells were not altered in Dmab-treated patients, while they dramatically decreased in Zol-treated patients. In the MRONJ control group, Zol-ONJ patients showed a reduction in activated T cells and γδT cells compared to Dmab-ONJ patients. Dmab was less immunosuppressive than Zol, not affecting γδT cells and increasing activated T cells.

## 1. Introduction

Medication-related ONJ (MRONJ) is a multifactorial disease [1,2], occurring especially in the jaws, often at teeth extraction sites, in patients exposed to anti-resorptive drugs (ARDs) [3,4], and in a smaller measure to cyclin inhibitors and antiangiogenic drugs [5]. Indeed, recently, it has also been found that there is a correlation between MRONJ and antiangiogenic drugs, which induce microcirculation dysfunction, which contributes, in turn, to MRONJ [6,7]. Another risk factor contributing to ONJ onset is represented by the mechanical loading to which jaws are subjected, leading to a high bone turnover rate [8,9].

The role of bacterial infections in the pathogenesis of ONJ is also a debated point; indeed, commensal oral microbiota can be more or less promoting MRONJ [10,11,12,13]. At the early stage of MRONJ development, the commensal microbiome plays a protective role, but long-term treatment with antibiotics reduces or depletes it, favoring MRONJ. Thus, Ewald et al. conclude that a balance between reducing pathological bacteria and preserving the indigenous microbiome is a relevant factor in MRONJ onset [10].

The causal role of ARDs has been proven, and particularly, the time of exposure, cumulative dose, and administration intensity increase MRONJ risk [5,14,15,16]. Cancer patients with bone metastases who are treated with ARDs such as bisphosphonates (BPs) and the receptor activator of nuclear factor κB ligand (RANKL) antibody (Dmab) are more susceptible to developing MRONJ [6,14]. ARDs inhibit osteoclast (OC) function with a consequent suppression of bone turnover, normally due to the balanced activities of OC and osteoblast (OB) [17]. Both N-BPs and Dmab affect the immune system of MRONJ patients [18,19,20] by hindering their capability to respond properly to immunological stress [18,21,22]. According to Soma et al., the administration of N-BPs is associated with a cytokine storm showing a strong inflammatory response, which can be prevented using anti-inflammatory cytokines such as anti-tumor necrosis factor-α (TNFα) and anti-interleukin 6 (IL-6) [3]. Indeed, in a murine model, the administration of either anti-inflammatory or antibiotic drugs significantly blocked Zol-induced ONJ following tooth extraction [23]. The role of different immune cell subsets such as γδ T cells [19], T helper 17 (Th17), T regulatory (T reg) cells [24], natural killer (NK) cells [25], dendritic cells (DCs) [26], neutrophils [27], and macrophages [24,28] has been proven in the pathogenesis of MRONJ. Nonetheless, most of the studies concern the effect of Zol on different immune cell subsets, and only a few works are focused on Dmab action, as we recently reviewed [29].

A fundamental role in the pathogenesis of MRONJ is played by myeloid cells in the oral barrier tissue, where they are recruited, and they guide wound healing in the injured barrier. Zol directly affects myeloid cells, reducing the healing process, as demonstrated in a mouse model where Zol administration, followed by tooth extraction, caused osteonecrotic lesions that were dependent on an increased localization of Ly6G+/Gr1+ myeloid cells in the oral barrier [30].

Zol also modulates γδ T cells, inducing the accumulation of isopentenyl pyrophosphate (IPP), a metabolite in the mevalonate pathway of the cholesterol metabolism [31,32], which initially stimulates γδ T cells but later induces their reduction [33]. N-BPs also induce a high release of reactive oxygen species from neutrophils, which leads to the loss of γδ T cells [27]. In a mouse model, Zol modulates the Treg/Th17 ratio, increasing Th17 and reducing Treg cells, which leads to an inflammatory condition promoting ONJ [24].

In the present work, we studied a small cohort of bone metastatic breast cancer patients who started ARD treatment for bone lesions to shed light on the effects of ARDs on circulating immune cell subsets and on their general activation state during the first 18 months of treatment. We monitored helper T cells (CD3+CD4+Th), cytotoxic T cells (CD3+CD8+Tc), γδ T cells, and osteoclast precursors (OCPs), pursuing a twofold aim: (a) to understand whether Zol and Dmab could differently affect the level and the activation state of these different immune cell subsets and (b) to evaluate whether these changes could be a predictive measure of the risk of developing MRONJ in the first 18 months of treatment.

## 2. Material and Methods

### 2.1. Patients and Study Design

Ethical approval for the present study was requested and obtained from the Institutional Review Board of the Città della Salute e della Scienza di Torino (IRB number: ID-552651). All patients signed their written informed consent to participate in this prospective study at the Oral Surgery Department of the Dental School of Turin. The characteristics of patients are reported in Figure 1. The study group was composed of 10 patients affected by breast cancer treated with cyclin inhibitors who developed bone metastases (58.5 ± 11.2; mean age ± SD), not assuming glucocorticoids or other drugs affected the immune system or bone turnover. Before starting ARDs to treat bone metastases, these patients were subjected to routine oral cavity examinations to exclude the potential evident risks of developing osteonecrosis soon after ARD intake. Five patients were treated with Zol and 5 with Dmab, and their blood samples were taken at 0 (T0), 6 (T6), 12 (T2), and 18 (T18) months to perform analysis on peripheral blood mononuclear cells (PBMCs) and sera. At every time point, patients were interviewed, and their clinical records were assessed to ensure that the anticancer treatment conditions remained unaltered for a long time. Two control groups were utilized, that is, (a) 8 patients with neo-diagnosis of MRONJ [4 patients with Dmab-induced ONJ (29 ± 11.3, mean months from the onset of ONJ ± SD) and 4 with Zol-induced ONJ (34 ± 16.2)] to compare their PBMCs with the study group’s ones and (b) 5 patients with neo-diagnosis of breast cancer, who had not received any treatment, to exclude that the cyclin inhibitor treatment affects the levels of cytokines in sera.

### 2.2. Flow Cytometry

At 0, 6, 12, and 18 months, PBMCs were collected, and the following immune subsets of cells were analyzed: CD4+ Th, CD8+ Tc, γδ T cells, and OCPs. On Th and Tc cells, we also evaluated the expression of two activation markers, namely, CD69 and CD25. We used the anti-human antibodies CD3 FITC, CD4 APC, PeRCP CD8 PerCP, γδ TCR PE, CD11b APC, CD51-61 PE (Miltenyi Biotec, Bergisch Gladbach, Germany), CD25 FITC, CD69 PE (BioLegend, SanDiego, CA, USA), and CD14 FITC (Thermo Fisher Scientific, Waltham, MA, USA), and related isotype and unstained controls. Appropriate controls were used to determine optimal voltage settings and electronic subtraction for the spectral fluorescence overlap correction. Samples were analyzed by the MACsQuant10 instrument and elaborated by MACs quantify software (Miltenyi Biotec, Bergisch Gladbach, Germany). Data represent a percentage of positive cells, determined by subtracting the percentage value of the appropriate isotype and unstained controls from each sample.

### 2.3. Cell Cultures

PBMCs were isolated from patients’ peripheral blood after centrifugation over a density gradient, Lymphoprep (Nycomed Pharma, Zürich, Switzerland). Cells were cultured in α-MEM supplemented with FBS, 100 IU/mL penicillin, and 100 μg/mL streptomycin and then counted and plated (Thermo Fisher Scientific, Waltham, MA, USA). To obtain fully differentiated human OCs, PBMCs were cultured for 15 days. As a positive control, PBMCs were also cultured in the presence of recombinant human M-CSF (25 ng/mL) and RANKL (30 ng/mL) (Peprotech, London, UK). Culture medium was refreshed every 5 days, either supplemented or not, with the factors reported above. At the end of the culture period, mature OCs were stained by the tartrate-resistant acid phosphatase (TRAP) kit (Cosmo Bio Co., Ltd. Tokyo, Japan) and identified as TRAP-positive, multinucleated cells containing three or more nuclei. OCs were counted using a microscope, and the presence of spontaneous osteoclastogenesis was considered significant when the cut-off number of 70 cells was reached or overcome, according to previously published works (16).

### 2.4. Cytokine Dosage

We collected patients’ sera from blood samples after centrifugation at 3000 rpm for 10 min at each time point (0, 6, 12, and 18 months), and we analyzed 12 pro-inflammatory cytokines according to the MACSPlex Cytokine 12 Kit (Miltenyi Biotec, Bergisch Gladbach, Germany), which is designed for determining concentrations of multiple soluble analytes (IL-2, IL-4, IL-5, IL-6, IL-9, IL-10, IL-12, IL-17, IFN-α, IFN-γ, TNF-α, and GM-CSF) in a single sample. The analysis was based on MACSPlex capture beads, which were a cocktail of different bead populations labeled with various fluorocromes. Each bead population was coated with a specific antibody, reacting with one of the analytes within the sample. A standard curve was obtained starting from the dilution of the standard concentrate provided with the kit, and it was utilized to quantify the analytes in the tested sera samples. Indeed, samples containing unknown levels of analytes were incubated with antibody-coated MACSPlex capture beads to promote binding with specific antibodies. Then a detection reagent containing a cocktail of fluorescent conjugated antibodies specific to the analytes was added. The different fluorescences corresponding to the analytes were identified using standard flow cytometry techniques. Samples were analyzed by the MACsQuant10 instrument and elaborated by the MACs quantify software (Miltenyi Biotec, Bergisch Gladbach, Germany).

### 2.5. Statistical Analyses

STATA software (version 17.0; StataCorp, College Station, TX, USA) was used for statistical analysis. Data were subjected to a Student’s *t*-test to evaluate statistically significant differences between the variables in the denosumab and zoledronate groups. A one-way ANOVA with Dunnet’s multiple comparison test was performed to assess differences in the serum values of the 12 cytokines in the two groups at different observational times. Statistical significance was considered at *p* < 0.05.

## 3. Results

### 3.1. Dmab-Treated Patients Show an Increased Level of CD69, a T Cell Activation Marker

The analysis of CD4+ T cells showed that their level did not significantly change during the time or due to the different anti-resorptive treatments (Figure 2A). Conversely, cytotoxic CD8+ T cells remained stable in Dmab-treated patients, while they decreased in Zol-treated patients, with a significant difference at T18 compared to T0 (patients before starting the ARD treatment) (Figure 2B). Looking at the expression of activation markers, we observed in Dmab-treated patients a significant increase in CD4/CD69+ cells (3.4 ± 1.4 vs. 0.7 ± 0.6; 3.4 ± 1.4 vs. 1.3 ± 0.8% of positive cells ± SD, ** *p* < 0.01 and * *p* < 0.05) at T18 compared to T0 and T6, respectively (Figure 2C). CD8+ Tc cells expressed significantly more CD69 at T18 than at T0 (4.6 ± 0.5 vs. 0.7 ± 0.3, *p* < 0.01) (Figure 2D).

We also evaluated the expression of CD25 on both CD4+ and CD8+ T cells, which was not significantly different in the two groups at any time (Appendix A).

To investigate whether the above-described trends of T cell subsets and their different behaviors according to ZOL and Dmab treatment were comparable or not with the status of T cells in patients who developed MRONJ, we performed the same analysis on the latter. In the group of patients with MRONJ induced by Zol, we observed a higher level of CD4+ and CD8+ cells and a decreased level of CD4/CD69+ T cells compared to patients with Dmab-induced ONJ, *p* < 0.05 (Figure 3).

### 3.2. Dmab-Treated Patients Show a Normal Level of Circulating γδ T Cells

We monitored γδ T cells due to their reported role in maintaining bone oral cavity homeostasis, observing that, in Dmab-treated patients, circulating γδ T cells remained constant for 12 months, and then an increasing trend was evident at T18. γδ T cell level dramatically decreased at T6 after starting Zol treatment (2.4 ± 1.9 vs. 0.5 ± 0.6, *p* < 0.05) and remained constantly low during the rest of the observational period (Figure 4A). A lower level of γδ T cells was also detected in patients who developed Zol-induced ONJ compared to patients with Dmab-induced ONJ (Figure 4B).

### 3.3. Circulating OCPs and OC Formation Decrease after 12 Months of Treatment with ARDs

The level of circulating OCPs is generally associated with the OC number and activity; thus, it is augmented in diseases characterized by increased bone resorption, such as bone metastases. For both Dmab- and Zol-treated patients, circulating OCPs were stable for the first 6 months, while a decrease occurred later at T12 (Figure 5A). To verify whether ARD treatment may affect OC formation, we performed an osteoclastogenesis in vitro assay, showing that at T12, the number of TRAP+ OCs was reduced compared to that at T0 (75 ± 31 vs. 112 ± 46, mean OC number ± SD). Even though the difference was not significant, the morphology showed large and multinucleated TRAP+ OCs at T0 (Figure 5B,D) and small TRAP+ OCs (with no more than 3 or 4 nuclei) at T12 in both the Zol- and Dmab-treated patients (Figure 5C,E).

### 3.4. Cytokines Modulation in Sera According to ARDs

The level of pro-inflammatory cytokines was detected in the sera of patients with a neo-diagnosis of cancer (control group) before they started anticancer treatment to exclude bias in the cytokine levels dependent on therapy. We could not detect any significant difference in circulating cytokines among patients in the control group and our series of Dmab- or Zol-treated patients (Appendix A). By analyzing the single group of treatment (i.e., Dmab and Zol), we did not detect any variation in cytokine level during the monitoring time, except for IFN-γ, which became undetectable in Zol-treated patients at T12 and T18 (Table 1).

Conversely, we detected differences between the two groups of patients at specific time points. IFN-α was higher in Dmab-treated patients than in Zol ones at T6; GM-CSF, IL-5, and IL-10 were significantly increased at T6 and T12; and IL-12 and TNF-α were higher at T6, T12, and T18 (Figure 6). IL-4, IL-6, and IL-17 did not show significant variations between the two groups. IL-2 and IL-9 were completely nondetectable (Table 1).

## 4. Discussion

The role of immune responses and inflammation in the onset and/or progression of MRONJ has become more and more evident since the first papers on the matter [34,35]. Here, in 10 bone metastatic breast cancer patients treated with either Dmab or Zol, we investigated and compared the possible effects elicited on different circulating immune subsets to evaluate whether any changes in circulating immune subsets take place during the first 18 months of ARD treatment and whether they could be predictive of MRONJ development.

During the observational period, we did not register any significant variation in the level of circulating CD4+ Th cells with both the ARDs, although differences were detected for cytokines regulating Th cells. In particular, the serum levels of TNF-α and IL-12 were higher in Dmab-treated patients than in Zol ones. Notably, a main effect of IL-12 is the induction of IFN-γ, which, in turn, stimulates Th1 cells endowed with anti-inflammatory and anti-tumor functions [36,37]. In Dmab-treated patients, the levels of IFN-γ remained stable, as did the Th cells. Conversely, in Zol-treated patients, who also had lower IL-12 levels, IFN-γ decreased dramatically and became undetectable at T12 and T18. Moreover, we detected a significant increase in GM-CSF, IL-5, and IL-10 levels in the sera of Dmab-treated patients at T6 and T12. According to the literature, Zol and the RANKL inhibitor OPG blocked GM-CSF-induced tumor growth in bone, but this did not affect the capability of GM-CSF to reverse chemotherapy-induced leukopenia [38]. Thus, in Dmab-treated patients, the increase in GM-CSF could help maintain the correct pool of leukocytes and could also depend on activated Th1, as previously reported [39]. Furthermore, GM-CSF promotes the differentiation of myeloid cells, which are also the main producers of IL-10, an anti-inflammatory cytokine [40].

In the study group, Zol may be responsible for the reduction in circulating CD8+Tc cells, since we observed that their level progressively decreased in circulation until 18 months, while in Dmab-treated patients, they remained stable. Conversely, in Zol-induced ONJ patients, CD8+Tc cells significantly increased compared to those in Dmab-ONJ patients, but they did not express activation markers, thus suggesting an immunosuppressive action of Zol on this cell subset. Looking at the expression of the activating marker CD69 on CD4+ Th cells, we observed that Dmab induced a progressive increase in CD69+ Th cells during the time, while in CD69+ Tc cells, the increment was significant only at T18. This observation may be of interest in light of the increase in CD69+ Th cells that was detected in Dmab-induced ONJ patients, suggesting perhaps a contributing role of Dmab treatment in maintaining such an increase in these subjects. On the contrary, Zol did not affect CD69+ Th cells, suggesting that Dmab might allow the maintenance of circulating activated immune subsets, which deserves further investigation. Indeed, CD69 is not only an activation marker but also regulates the retention of cells on which it is expressed, and it is involved in the pathogenesis of inflammatory disorders, irrespective of the type of inflammation; thus, activated T cells are recruited into the inflamed tissue through the interaction of CD69 with its ligands [41,42].

The role of γδ T cells in the pathogenesis of MRONJ is debated, even though γδ T cells are known to be fundamental for fracture healing, to which tooth extraction can be compared. An increased level of γδ T cells in MRONJ patients’ tissue and in murine models has been reported [43]. Both γδ T null and wild-type mice treated with Zol developed osteonecrosis, while after injecting human γδ T cells into Zol-immunodeficient (Rag2−/−) mice, osteonecrosis was not induced [19]. Looking at the mechanisms underlying MRONJ, it has been shown that activated γδ T cells produced Semaforin-4D (Sema4D), which is implicated in the pathogenesis of T cell-mediated inflammatory diseases [44], and it stimulates pro-inflammatory cytokine production [43]. Here, we reported a drop in the number of γδ T cells after 6 months of Zol treatment, according to the literature data [33,45]. This drop can be explained by supposing that N-BPs cause the accumulation of a metabolite in the mevalonate pathway, isopentenyl pyrophosphate (IPP) [31,32,46], which is an antigen identified by γδ T cells that initially stimulates them but later induces their reduction through a mechanism of activation-induced cell death commonly used by the immune system to maintain homeostasis [33,47,48]. Another hypothesis explains the loss of γδ T cells with the high release of reactive oxygen species from neutrophils taking up N-BPs, which results in γδ T cell suppression [22,27]. Interestingly, our Dmab-treated patients did not show a decrease in γδ T cells differently from Zol ones, and looking at Dmab-induced ONJ patients, they had higher levels of γδ T cells compared to Zol-induced ONJ ones, suggesting that Dmab and Zol act in different ways on γδ T cells. Dmab does not interfere with the mevalonate pathway, but it is an anti-RANKL, so it can act on T cell activation and on thymic selection during the development of γδ T cell progenitors [49]. Kalyan et al. hypothesized that Dmab lowered the immune defenses by reducing T cell development [50]. Indeed, the RANK-RANKL system is a fundamental bridge between the immune system and bone [51,52], and a gene expression analysis of MRONJ patients deficient for γδ T cells showed a low expression of immune factors such as RANKL, RANK, TNF-α, and GM-CSF [21].

In our series, only one Dmab patient experienced MRONJ after 18 months of treatment, but his immune subsets remained stable, suggesting that the state of circulating cells likely does not reflect the oral cavity damage induced by MRONJ. Altogether, our data suggest that γδ T cells contribute to ONJ pathogenesis, but they do not mediate the core mechanism for ONJ onset, consistent with Park et al. [19].

Since γδ T cells are activated by OC deriving from peripheral blood monocytes [53], we assessed the level of circulating OCPs, showing that they were stable for the first 6 months, while a decrease started later at 12 months for both the ARDs. We also evaluated the capability of OCPs to differentiate spontaneously into OCs, which is a characteristic of pathologies associated with increased bone resorption, such as bone metastases [54,55]. As expected, we observed marked spontaneous osteoclastogenesis in patients before starting ARDs, but after 12 months, the number of OCs reduced, confirming that ARDs were effective in blocking OC activity according to the reduction in circulating OCPs.

This study has some limitations, such as the small number of enrolled patients, mainly due to the fact that cancer patients are often subjected to changes or therapy adjustments, and thus, it is difficult to obtain a clean series of patients. Moreover, we did not investigate the gut microbiota of breast cancer patients, even though accumulating evidence suggests that it can trigger both innate and adaptive immune responses through immunomodulatory cytokines and chemokines affecting tumor progression and response to anticancer therapies [56,57]. Indeed, cyclin inhibitor treatment efficacy can be improved by probiotics [57], and both dysbiosis and estrogen inhibition are associated with a reduction in γδ T cells in terms of number and activity. Thus, we cannot exclude the involvement of dysfunction in the gut microbiota in our cohort. In our clinical experience, Dmab-treated patients develop osteonecrosis before Zol ones, as is also evident by the different mean times of MRONJ development, dependent on Dmab or Zol treatment, observed in our MRONJ control group. In the present research, only one patient developed MRONJ, and the onset occurred at T18, a time that is consistent with the range covered by Dmab-induced MRONJ patients. Another interesting clinical observation is that patients treated with Dmab usually display less severe osteonecrotic features with faster healing lesions than Zol ones. We can speculate that this could depend on the lower immunosuppressive state induced by Dmab compared to Zol, also considering that Zol-treated patients completely lose the γδ T cell subset, which is paramount for the maintenance of oral cavity bone homeostasis.

## 5. Conclusions

MRONJ occurs commonly after dental extraction, which requires a healing process similar to bone regeneration. A coupled process of bone resorption and formation is also physiologically regulated by the immune system and guarantees effective bone remodeling. Thus, treatments that interfere with bone remodeling, affecting the immune–bone interface, should be carefully evaluated. ARDs can affect circulating immune cell subsets, altering their activation state and promoting an inflammatory microenvironment responsible for immunosuppression, which eventually does not sustain the correct bone remodeling. According to our results, ARDs differ in their capacity to modulate the immune system, potentially explaining the different clinical evolution of ONJ induced by Dmab or Zol. Indeed, in our clinical observation, patients treated with Dmab usually display lesions characterized by less severe osteonecrotic features with faster healing lesions than Zol ones, which might depend on the maintenance of a more active state of the immune system in Dmab-treated patients.

## Figures and Tables

**Figure 1 cells-12-02430-f001:**
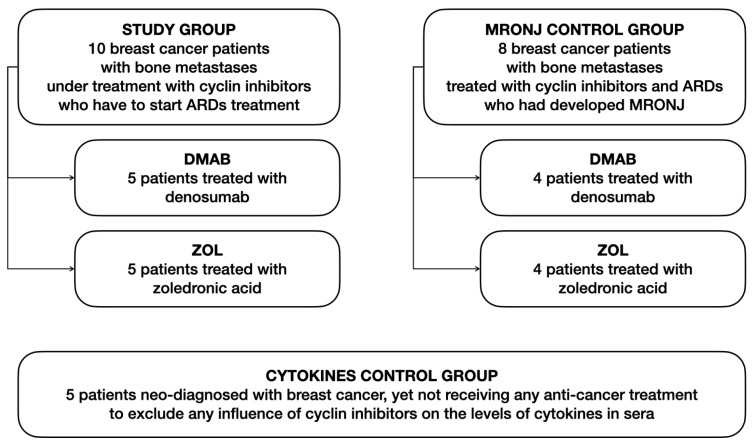
Diagram depicting the study design and patient groups involved in the study.

**Figure 2 cells-12-02430-f002:**
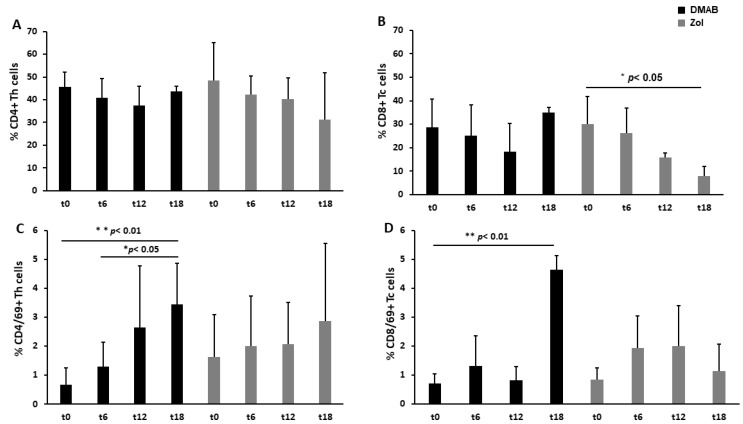
Circulating T cell phenotypes in the two groups of study. CD4+ Th cells were stable in both groups of treatment and at different time points (**A**). CD8+ Tc significantly decreased in Zol-treated patients at 18 months, *p* < 0.05 (**B**). The subsets of activated CD69+ cells progressively increased in Th and Tc cells of Dmab-treated patients at 18 months (**C**,**D**).

**Figure 3 cells-12-02430-f003:**
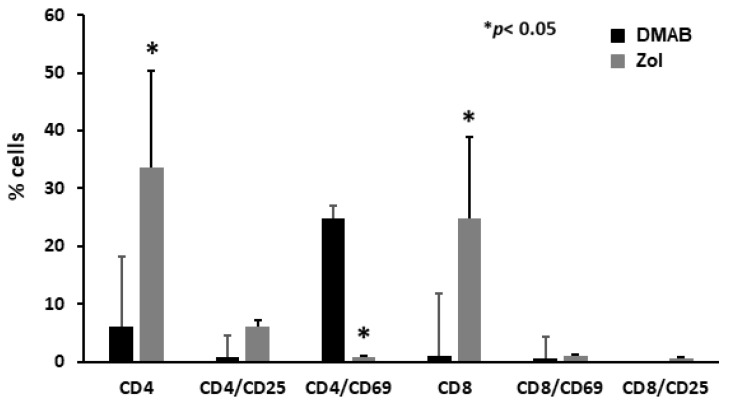
Circulating T cell subsets in MRONJ patients. Circulating Th CD4+ and Tc CD8+ cells were increased in Zol-treated patients compared to Dmab ones (*p* < 0.05). Conversely, activated CD4/CD69+ T cells were significantly decreased only in Zol-treated patients (*p* < 0.05).

**Figure 4 cells-12-02430-f004:**
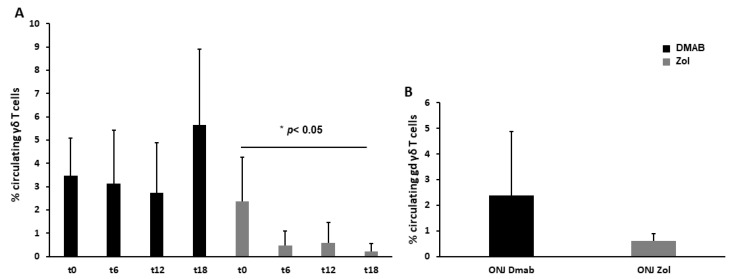
Dmab and Zol effects on γδ T cells. In Dmab-treated patients, the level of circulating γδ T cells remained constant with an increasing trend at 18 months, while it decreased soon after Zol treatment (**A**). In the group of patients with ONJ, γδ T cells were lower in patients with Zol-induced ONJ (**B**).

**Figure 5 cells-12-02430-f005:**
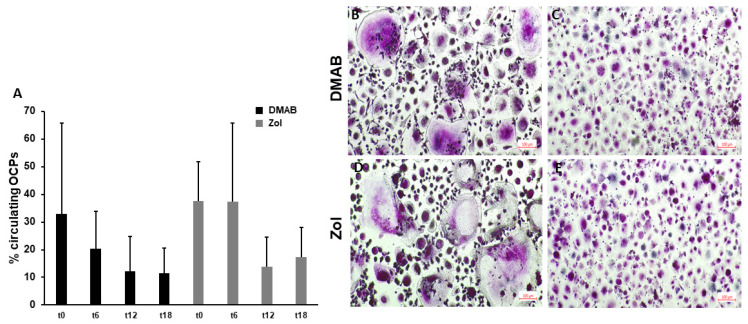
Circulating OCPs and OC formation. The number of circulating OCPs decreased during treatment with ARDs, without significant differences (**A**). Osteoclastogenesis in vitro at T0 (**B**,**D**) was reduced, with fewer large multinucleated TRAP+ OCs (**C**,**E**) after 12 months of treatment with both ARDs.

**Figure 6 cells-12-02430-f006:**
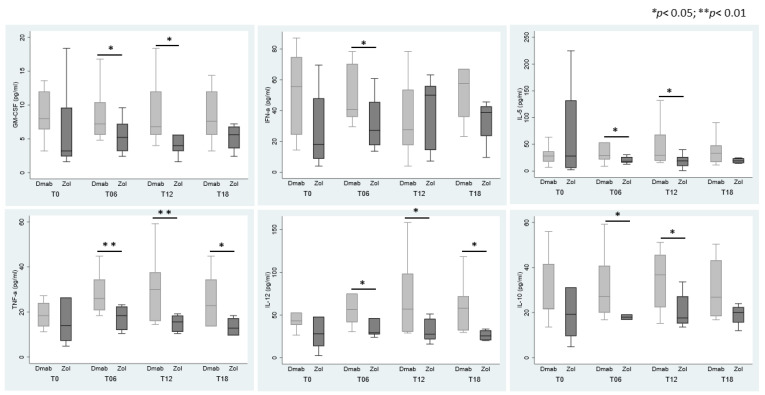
Cytokine expression in sera. Graphs show the serum levels of cytokines, which were significantly increased in Dmab-treated patients compared to Zol ones at the different time points.

**Table 1 cells-12-02430-t001:** Cytokines serum levels. The mean ± SD (pg/mL) of the serum levels of cytokines analyzed in the patient’s sera at the different time points are reported.

	DMAB				ZOL			
	T0	T6	T12	T18	T0	T6	T12	T18
**GM-CSF**	8.6 ± 1.3	8.6 ± 1.4	8.6 ± 1.4	8.4 ± 1.8	6.8 ± 2.0	5.5 ± 1.1	4.0 ± 0.6	5.2 ± 1.1
**IFNα**	51.4 ± 9.8	50.0 ± 6.2	35.8 ± 7.6	59.9 ± 13.4	29.7 ± 7.9	32.0 ± 7.4	40.1 ± 9.5	33.2 ± 8.0
**IFNγ**	13.4 ± 7.0	32.9 ± 13.4	41.9 ± 14.7	24.3 ± 20.7	114.9 ± 55.1	10.4 ± 7.2	not detectable	not detectable
**IL-4**	395.8 ± 43.1	419.8 ± 55.1	419.0 ± 53.0	467.2 ± 70.9	328.6 ± 34.7	412.1 ± 25.7	446.9 ± 24.4	380.6 ± 43.9
**IL-5**	29.6 ± 6.1	45.0 ± 12.1	49.8 ± 12.8	38.8 ± 11.6	69.9 ± 33.8	20.3 ± 2.7	19.2 ± 5.5	19.4 ± 2.7
**IL-6**	9.8 ± 0.9	10.6 ± 1.0	11.8 ± 2.3	9.6 ± 2.5	8.1 ± 1.3	10.0 ± 1.8	9.5 ± 1.4	10.2 ± 2.0
**IL-10**	30.5 ± 4.9	32.8 ± 4.9	34.0 ± 4.2	30.4 ± 5.9	32.6 ± 10.8	20.1 ± 4.5	20.8 ± 3.2	19.0 ± 2.5
**IL-12**	52.5 ± 9.0	69.2 ± 13.5	73.6 ± 16.2	61.3 ± 13.3	72.2 ± 32.6	38.7 ± 8.1	31.6 ± 5.7	26.2 ± 3.4
**IL-17A**	17.9 ± 4.9	28.3 ± 5.6	27.3 ± 7.1	29.6 ± 13.5	35.2 ± 11.5	19.7 ± 5.7	13.2 ± 3.1	16.4 ± 6.1
**TNFα**	21.8 ± 3.7	27.8 ± 2.8	29.6 ± 4.4	25.3 ± 5.1	23.8 ± 7.7	17.5 ± 2.2	15.1 ± 1.5	13.4 ± 2.2

## Data Availability

The data presented in this study are available on request from the corresponding author. The data are not publicly available due to privacy.

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
