# Peer review of "Denosumab and Zoledronic Acid Differently Affect Circulating Immune Subsets: A Possible Role in the Onset of MRONJ"

_cells, 2023, doi:10.3390/cells12202430_

Round 1

Reviewer 1 Report

The authors have discussed the response of anti-resorption drugs Zoledronic acid and Denosumab to the levels of circulating immune cell subsets, and the risk prediction for developing Medication-related ONJ (MRONJ).

However, the manuscript requires MAJOR revisions-

1. The Abstract should be structured, including a brief background, Aims and objectives, Study results, and conclusion.

2. The keywords should be arranged alphabetically

3. In the Introduction, a diagrammatic representation of the role of Anti-resorptive drugs (ARDs) in development of MRONJ would be appreciated. This will facilitate better understanding

4. Please mention the Study design in methodology section?

5. 

6. Line no. 84, Two control group were foreseen as follows. Please rephrase this.

7. Fig. 1 Legend- The diagram resumes the characteristic of study and control groups should be rephrased.

8. In the discussion, please Rephrase the lines 223-227 as they are difficult to comprehend

9.  Please mention the strength and limitation of the study

10. English language needs serious concerns

English language needs serious concerns

Author Response

  1. The Abstract should be structured, including a brief background, Aims and objectives, Study results, and conclusion.

Thank you for your suggestion, but the abstract has been written according to the journal instructions:

 “The abstract should be a total of about 200 words maximum. The abstract should be a single paragraph and should follow the style of structured abstracts, but without headings: 1) Background: Place the question addressed in a broad context and highlight the purpose of the study; 2) Methods: Describe briefly the main methods or treatments applied. Include any relevant preregistration numbers, and species and strains of any animals used; 3) Results: Summarize the article's main findings; and 4) Conclusion: Indicate the main conclusions or interpretations”

  1. The keywords should be arranged alphabetically

Done

  1.  In the Introduction, a diagrammatic representation of the role of Anti-resorptive drugs (ARDs) in development of MRONJ would be appreciated. This will facilitate better understanding

Thank you for your kind suggestion, but we think that representing the role of the different ARDs in the pathogenesis of ONJ may transcend the aim of our paper. Neither is it easy to summarize such role in a diagram according to the complexity of the matter, which is still to be fully elucidated.

  1. Please mention the Study design in methodology section?

Done

  1. Line no. 84, Two control group were foreseen as follows. Please rephrase this.

Done

  1. 1 Legend- The diagram resumes the characteristic of study and control groups should be rephrased.

Done

  1. In the discussion, please Rephrase the lines 223-227 as they are difficult to comprehend

Done

  1. Please mention the strength and limitation of the study

Done

  1. English language needs serious concerns

      An English mother language revised the manuscript

Reviewer 2 Report

The overall information in this article is quite interesting.

1 It would be better to talk about the collateral effects of chemotherapy given to the patients

2 Breast cancer has many particular aspects that eventually interfere, interface, and influence with gamma delta T-cells. Estrogen inhibitors have been known to reduce the availability of local mesenchymal stem cells and thus osteoblasts; the low level of estrogen is usually associated with a low level of serotonin and a high level of prolactin, a picture which suggests a gut dysbiosis'

3 Dysbiotic environments usually see the presence of low reactive gamma/delta T-cells considered as a bridge between adaptive immunity and innate immunity.

4 gamma/delta T-cells rapid activation and effector functions, with a capacity for cytotoxic anti-tumor responses and production of inflammatory cytokines such as IFN-γ or IL-17 strictly depend on well-balanced gut microbiota. These observations also have significant implications for anti-tumor therapy and vaccination, suggesting that the communication between gamma/delta T cells and the microbiota involves soluble mediators (microbiota-derived metabolites) that influence various functions of T cells.

5 In addition it should be stressed more the relationship between estrogen, chemotherapy, and bone decay

minor revision needed

Author Response

We thank the reviewer for his suggestions, that allowed us to improve the manuscript. In particular, we discussed the role of oral, gut microbiota and immune system both in introduction and in discussion.

About the role of chemotherapy, we did not speak about it, because patients enrolled in the study were treated with cyclin inhibitors, but not with chemoterapeutic agents.

Round 2

Reviewer 1 Report

Dear Authors

Thank you for incorporating most of the suggestions in the revised manuscript.

Wishing you all the best

Minor editing of the English language is required.